



# Validation of initial observation from the first space-borne high spectral resolution lidar with ground-based lidar network

Qiantao Liu [1], Zhongwei Huang [1,2], Jiqiao Liu [3,4], Weibiao Chen [3,4], Qingqing Dong [1], Songhua Wu [5,6], Guangyao Dai [5], Meishi Li [1], Wuren Li [2], Ze Li [2], Xiaodong Song [2], and Yuan Xie [3]

[1] Key Laboratory for Semi-Arid Climate Change of the Ministry of Education, College of Atmospheric Sciences, Lanzhou University, Lanzhou, 730000, China

[2] Collaborative Innovation Center for Western Ecological Safety, Lanzhou University, Lanzhou, 730000, China

[3] Shanghai Institute of Optics and Fine Mechanics, Chinese Academy of Sciences, Shanghai, 201800, China

[4] Center of Materials Science and Optoelectronics Engineering, University of Chinese Academy of Sciences, Beijing, 100049, China

[5] College of Marine Technology, Faculty of Information Science and Engineering, Ocean University of China, Qingdao, 266100, China

[6] Laoshan Laboratory, Qingdao, 266237, China

*Correspondence to*: Zhongwei Huang (huangzhongwei@lzu.edu.cn)

**Abstract.** On April 16, 2022, China has successfully launched the world's first space-borne high spectral resolution lidar (HSRL), which is called the Aerosol Carbon Detection Lidar (ACDL) onboard the DQ-1 satellite. The ACDL is expected to precisely detect the three-dimensional distribution of aerosol and cloud globally with high spatial-temporal resolutions. To assess the performance of the newly launched satellite lidar, the ACDL retrieved observations were compared with ground-based lidar measurements of atmospheric aerosol and cloud over northwest China from May to July 2022 by the exploitation of the Belt and Road lidar network (BR-lidarnet) initiated by Lanzhou University of China, as well as CALIPSO lidar observations. A total of six cases at daytime and nighttime, including clear days, dust events, and cloudy conditions, were selected for further analysis. Moreover, profiles of total attenuated backscatter coefficient (TABC) and volume depolarization ratio (VDR) at 532 nm measured by the ACDL, CALIPSO lidar and ground-based lidar are compared in detail. Comparison between the 532 nm extinction coefficient and lidar ratio obtained from ACDL HSRL retrieval and the Raman retrieval results obtained from BR-lidarnet. The achieved results revealed that the ACDL observations were in good agreement with the ground-based lidar measurements during dust events with a relative deviation of about -10.5 ± 25.4 % for the TABC and -6.0 ± 38.5 % for the VDR. Additionally, the heights of cloud top and bottom from these two measurements were well matched and comparable. Compared with the observation of CALIPSO, it also shows high consistency. This study proves that the ACDL provides reliable observations of aerosol and cloud in the presence of various climatic conditions, which helps to further evaluate the impacts of aerosol on climate and environment as well as the ecosystem in the future.



# 1 Introduction

The ACDL is the first space-borne HSRL, which was successfully launched on 16 April 2022 onboard the Atmospheric Environmental Monitoring Satellite (AEMS) also known as DaQi-1 (DQ-1). The ACDL is capable of providing a novel perspective on the spatial distribution of atmospheric aerosols and clouds across the globe. An iodine cell is employed to distinct Mie backscattering and Rayleigh backscattering signals from the atmosphere by the ACDL (Dong et al., 2018) so that aerosol and cloud extinction coefficients are precisely retrieved without assuming a lidar ratio (Comerón et al., 2017; Sugimoto and Huang, 2014). Therefore, the ACDL could remarkably enhance the understanding of the crucial roles of aerosols and clouds in global weather and climate change due to their radiative forcing of the Earth-atmosphere system (IPCC, 2013). Still, aerosol inhomogeneity and complex macro- and microphysical processes make them difficult to observe and are therefore the most uncertain part of the model. The exploitation of lidar as an effective tool to explore aerosols and clouds has been now extensively developed due to its active detection as well as its advantages in better penetration and temporal continuity (Comerón et al., 2017; Sugimoto and Huang, 2014; Huang et al., 2018; Liu et al., 2022; Huang et al., 2022). Thus, satellite-based lidar detection technology can be implemented to observe continuous spatial-temporal aerosol and cloud distribution around the world, thereby compensating for the deficiency in ground-based lidar detection and playing an increasingly prominent role in all current observation approaches (Huang et al., 2018; Liu et al., 2022; Huang et al., 2015; Sugimoto et al., 2012; Huang et al., 2010). The combination of ground observation and satellite observation can be used to evaluate the direct radiative forcing caused by aerosols (Hansell et al., 2010; Lakshmi et al., 2021), analyze the outbreaks of desert dust (Marcos et al., 2016). Long-term ground-based and space-borne lidar measurements could be used to understand the vertical and spatial distribution of aerosols (Gupta et al., 2021).

In order to ensure the accuracy and quality of satellite observation, satellite-based lidar data must be appropriately validated by ground-based lidar (Gimmestad et al., 2017). For instance, observations by the Cloud-Aerosol Lidar with Orthogonal Polarization (CALIOP) onboard the Cloud-Aerosol Lidar and Infrared Pathfinder Satellite Observations (CALIPSO) satellite have been globally validated. Chiang et al. (2011) reported the first inter-comparison of vertical profiles of aerosols and clouds derived from CALIPSO and ground-based lidar over Chung-Li, Taiwan. Many former investigations have employed ground-based lidar as a means of validating CALIPSO Level 1 products (Wu et al., 2011, 2014). An investigation by the European Aerosol Research LIDAR Network (EARLINET) of 40 CALIPSO overpassed cases was conducted to validate the initial CALIPSO backscattering attenuation coefficients (Mamouri et al., 2009). A comparison of CALIPSO and ground-based lidar subjected to various weather conditions was performed through a direct comparison of satellite and ground-based lidar observations (Tao et al., 2008). The CALIPSO Level 1 product was evaluated on the basis of one year of ground-based Raman lidar observations by comparing CALIPSO results at night and in the presence of clouds (Mona et al., 2009). A comparative verification of aerosol and cloud structure observed by CALIPSO lidar was achieved via ground-based dual-wavelength polarization lidar (Kim et al., 2008). Córdoba-Jabonero et al. (2017) used ground-based lidar combined with CALIPSO detection to study subtropical and polar cirrus clouds properties and compared the differences in their observations.





Furthermore, the level 2 extinction coefficient profile of CALIPSO has also been verified by using ground-based lidar
observations (Elina Giannakaki et al., 2011). CALIPSO version 2/version 3 daytime aerosol extinction product were accurately
evaluated based on a detailed multi-sensor, multi-platform (Kacenelenbogen et al., 2011). In addition, the accuracy and
uncertainty of aerosol products obtained by VIIRS through the deep blue algorithm are also compared and evaluated through
ground-based observations (He et al., 2021). The HSRL for detecting the optical properties of clouds and aerosols of ACDL
satellite has been evaluated through CALIPSO (Liu et al., 2019). Currently, another instrument carried on DQ-1 for detecting
carbon dioxide is being evaluated more frequently (Wang et al., 2020, 2022; Cao et al., 2022), but the ability of ACDL to
detect aerosol and cloud characteristics has not yet been evaluated using ground-based lidar systems.

The Belt and Road lidar network (BR-lidarnet) led by the Lanzhou University can provide real time data at multiple
wavelengths from ultraviolet to near-infrared (i.e., 355, 532, and 1064nm). Lidar network data are widely employed to validate
and assimilate dust transport models to assess the emission, transport, and deposition of dust (Sugimoto and Huang, 2014).
Huang et al. (2020) established a good relationship between the absorption coefficients of aerosols and the depolarization
ratios of dust aerosols at wavelengths of 532 nm and 355 nm, and in turn, a simple and reliable method was proposed to better
identify aerosol and cloud types (Qi et al., 2021). For the specific optical properties of dust aerosol, vertical distribution are
derived from observations at the Ruoqiang site in the lidar network combined with CALIPSO observations (Dong et al., 2022).
Likewise,  the vertical distribution of dust aerosols was obtained in the hinterland of the Taklimakan Desert based on
polarization Raman lidar observations at the Tazhong site in the lidar network (Zhang et al., 2022).

To verify the initial and novel retrievals of the ACDL, the vertical structure of aerosol and cloud near dust sources in northwest
China measured by the BR-lidarnet was implemented in the current work. The manuscript is structured as follow. The section
2 comprises with a brief introduction of the basic principles of the ACDL and ground-based lidar observation networks and
the methods exploited in this study that allow direct comparison of the two observations and comparison error assessment. In
Section 3, we present a comparison of the ACDL and ground-based lidar network observations in the presence of various
weather conditions. Finally, we conclude with a summary and discussion of the results in Section 4.

## 2 Lidar systems and methods

### 2.1 Space-borne ACDL

The ACDL, on board the satellite DQ-1, is primarily developed by the Shanghai Institute of Optics and Fine Mechanics, the
Chinese Academy of Sciences. The unique advantage of the ACDL is to precisely detect the global distribution of aerosols
and clouds at 532 nm with high spatio-temporal resolutions. By employing an iodine cell filter, Mie scattering and Rayleigh
scattering signals at 532 nm are separated (Dong et al., 2018). Additionally, polarization measurement at 532 nm is very
beneficial to distinguish non-spherical particles (dust and ice crystal) from spherical particles (water droplets and sea salt). In
addition, it is also detectable that Mie scattering signals of aerosol and cloud at the near-infrared wavelength (1064 nm) can
be employed to explore the size of particles combing with measurements at 532 nm. The spatial and temporal resolutions of


the ACDL observation are 24 meters and 1 minute, respectively. The main data products that will be provided by the ACDL in the near future are aerosol/cloud optical parameters, including the aerosol optical depth (AOD), cloud optical depth (COD), the height of aerosol/cloud layer, profiles of backscatter coefficient, extinction coefficient, and lidar ratio for aerosol and cloud. For the purpose of validation, the Level-2A product of the ACDL is employed, including the total attenuated backscattering coefficient (TABC) and volume depolarization ratio (VDR) at 532 nm using the high-gain channel. By utilizing the HSRL detection channel of ACDL, we can obtain extinction coefficient and lidar ratio at 532nm.

### 2.2 Ground-based lidar network (BR-lidarnet)

The ground-based lidar system developed by Lanzhou University is a dual-wavelength polarization-Raman lidar to detect tropospheric aerosol backscatter coefficients at 532 nm, 1064 nm, and volume depolarization ratio at 532 nm both in the daytime and nighttime. Additionally, backscattered Raman signal from the atmospheric $N_2$ molecules at 607 nm can be received at nighttime. The schematic representation of the lidar system exploited in the current investigation has been demonstrated in Figure 1. The lidar system employs an Nd: YAG laser equipped with a frequency-doubling crystal. The two laser beams at wavelengths of 1064 nm and 532 nm are collimated and amplified by beam expanders. A Cassegrain telescope is employed to receive backscattering signals from the atmosphere. The backscatter signal with a wavelength of 532 nm is divided into the parallel and perpendicular components by the polarizing beam splitters (PBS) and thus detected by the photomultiplier tubes (PMT), while the 1064 nm signal is detected by enhanced Si-Avalanche Photodiode (APD). The temporal and spatial resolutions of the observed data are 5 min and 3.75 m, respectively. Raw signals were pre-processed by background subtraction, range correction, polarization correction and overlap correction before further analysis (Wang et al., 2018; Huang et al., 2020).

### 2.3 CALIPSO Space-borne Lidar

The Cloud-Aerosol Lidar and Infrared Pathfinder Satellite Observation (CALIPSO) is designed to understand how clouds and aerosols influence earth's climate. On 28 April 2006, CALIPSO was launched as a new mission for earth science observation. CALIPSO is equipped with the cloud-aerosol lidar, known as CALIOP (Cloud-Aerosol Lidar with Orthogonal Polarization). The CALIOP system employs two-wavelength elastic backscatter lidar (Winker et al., 2009). Backscatter profiles could be obtained at 532 and 1064 nm, as well as two orthogonal (parallel and perpendicular) polarization components at 532 nm. A description of the CALIOP data products has already been provided elsewhere (Vaughan et al., 2004). The CALIPSO level 1 data product is used in this study. For a comparison with ACDL and ground-based lidar, an analysis at 532 nm was conducted.


## 2.4 Retrieval methodologies

### 2.4.1 Total attenuated backscatter coefficient (TABC)

For ground-based lidar systems, the elastic lidar equation is given by:

$$X(z) = P(z)z^2 = C[\beta_1(z) + \beta_2(z)]T_G{}^2(z) \tag{1}$$

where P(z) denotes the signal received by the ground-based lidar system from the altitude z, C represents the lidar system constant, and finally, $\beta_1(z)$, $\beta_2(z)$ in order are the aerosols and molecules backscatter coefficients at altitude z. The one-way transmission from lidar to altitude z is provided by:

$$T_G(z) = \exp\left\{-\int_0^z [\alpha_1(z') + \alpha_2(z')]\, dz'\right\} \tag{2}$$

where $\alpha_1(z)$ and $\alpha_2(z)$ represent the aerosols and molecules extinction coefficients at altitude z, respectively.

For the satellite-based ACDL, the observed backscatter is:

$$\beta_s{}'(z) = [\beta_1(z) + \beta_2(z)]T_s{}^2(z) \tag{3}$$

where $T_S(z) = \exp\left\{-\int_z^{z_s} [\alpha_1(z') + \alpha_2(z')]\, dz'\right\}$

in which $z_s$ denotes the space-borne lidar calibration altitude.

It is assumed that the directions of both ground-based and satellite-based lidars that emit laser pulses would be vertical. The above relations indicate that $T_S(z)/T_G(z)$ increases/decreases with the increase of altitude. Thus, the opposite laser emission directions of the satellite-based and ground-based lidars resulting the comparison between the two complicated.

In the present study, the attenuated backscatter observed by ground-based lidar is converted to downward attenuated backscatter (Tao et al., 2008), which in turn is utilized to directly compare the satellite-based and ground-based lidar observations. The conversion relation is stated as follows:

$$\beta_G{}'(z) = [\beta_1(z) + \beta_2(z)]\exp\left\{-2\int_z^{z_r} [\alpha_1(z') + \alpha_2(z')]\, dz'\right\} \tag{4}$$

where $z_r$ denotes the maximum detection height of the ground-based lidar. Few or no aerosols and few molecules are considered to exist between $z_r$ and $z_s$. Therefore, the function $\beta_G{}'(z)$ can be employed for comparison with $\beta_s{}'(z)$. The functions $\beta_2(z)$ and $\alpha_2(z)$ in above relations are calculated by Mie theory, $\beta_1(z)$ and $\alpha_1(z)$ are retrieved by the conventional Fernald approach (Fernald, 1984). In the retrieval calculation, it is assumed that the lidar ratio is a constant (i.e., does not vary with height), and set as 50 sr. An investigation of uncertainties at S1=50 sr revealed that the relative errors of the retrieved downward attenuated backscatter at altitudes above 5 km were less than 5%, whereas at altitudes below 5 km the errors were as large as 18% (Tao et al., 2008).

### 2.4.2 Volume depolarization ratio at 532 nm (VDR)

The volume depolarization ratio is evaluated via the following relation:

$$\delta_{532}(z) = \frac{\beta'_{532s}(z)}{\beta'_{532p}(z)} = \frac{\beta_{532s}(z)T_S^2(z,z_S)}{\beta_{532p}(z)T_S^2(z,z_S)} = \frac{\beta_{532s}(z)}{\beta_{532p}(z)} \tag{5}$$





where $\beta'_{532s}(z)$ and $\beta'_{532p}(z)$ are the perpendicular and parallel attenuated backscattering coefficients at 532 nm, respectively,

$\beta_{532s}(z)$ and $\beta_{532p}(z)$ are the perpendicular and parallel backscattering coefficients at 532 nm, respectively.

The ACDL's volume depolarization ratio could be directly compared with that measured by ground-based lidar because atmospheric transmittance is eliminated in Eq. (5).

### 2.4.3 Raman method retrieval of ground-based lidar

The aerosol extinction coefficient using Raman method (Ansmann et al., 1992) is calculated by:

$$\alpha_{\lambda_0}^{aer}(z) = \frac{\frac{d}{dz}ln\left[\frac{N_R(z)}{P_{\lambda_R}(z)z^2}\right]-\alpha_{\lambda_0}^{mol}(z)-\alpha_{\lambda_R}^{mol}(z)}{1+\left(\frac{\lambda_0}{\lambda_R}\right)^k} \qquad (6)$$

Where $N_R(z)$ is the atmospheric molecular number density of the Raman scattering, and k is the Angstrom exponent. $\alpha_{\lambda_0}^{mol}(z)$, $\alpha_{\lambda_R}^{mol}(z)$ and $\alpha_{\lambda_0}^{aer}(z)$ are the extinction coefficients. The superscripts $aer$ and $mol$ represent aerosol and molecular, respectively. The subscripts $\lambda_0$ (532 nm) and $\lambda_R$ (607 nm) represent the elastic backscattering wavelength and Raman scattering wavelength, respectively. $P_{\lambda_R}(z)$ is the Raman backscattering signal.

The aerosol backscatter coefficient by Raman lidar is calculated as follows:

$$\beta_{\lambda_0}^{aer}(z) + \beta_{\lambda_0}^{mol}(z) = \left[\beta_{\lambda_0}^{aer}(z_0) + \beta_{\lambda_0}^{mol}(z_0)\right] \times \frac{P_{\lambda_R}(z_0)P_{\lambda_0}(z)N_R(z)}{P_{\lambda_R}(z)P_{\lambda_0}(z_0)N_R(z_0)} \times \frac{\exp\left\{-\int_{z_0}^{z}\left[\alpha_{\lambda_R}^{aer}(\xi)+\alpha_{\lambda_R}^{aer}(\xi)\right]d\xi\right\}}{\exp\left\{-\int_{z_0}^{z}\left[\alpha_{\lambda_0}^{aer}(\xi)+\alpha_{\lambda_0}^{aer}(\xi)\right]d\xi\right\}} \qquad (7)$$

Where the reference height $z_0$ is chosen at the height where $\beta_{\lambda_0}^{mol}(z_0) \gg \beta_{\lambda_0}^{aer}(z_0)$, so that $\beta_{\lambda_0}^{mol}(z_0) + \beta_{\lambda_0}^{aer}(z_0) \approx \beta_{\lambda_0}^{mol}(z_0)$. Finally, the lidar ratio profile can be obtained from the profiles of $\alpha_{\lambda_0}^{aer}(z)$ and $\beta_{\lambda_0}^{aer}(z)$.

$$S_{\lambda_0}^{aer}(z) = \frac{\alpha_{\lambda_0}^{aer}(z)}{\beta_{\lambda_0}^{aer}(z)} \qquad (8)$$

### 170   2.4.4 High spectral resolution lidar (HSRL) retrieval method from ACDL

The lidar equations of HSRL are described as follows (Dong et al., 2018):

$$P_C^\perp(z) = \frac{P_0\eta_1A_rL}{z^2}(\beta_{mol}^\perp + \beta_{aer}^\perp)\exp\left(-2\int_0^z(\alpha_{mol}+\alpha_{aer})d\xi\right) \qquad (9)$$

$$P_C^\parallel(z) = \frac{P_0\eta_2A_rL}{z^2}(\beta_{mol}^\parallel + \beta_{aer}^\parallel)\exp\left(-2\int_0^z(\alpha_{mol}+\alpha_{aer})d\xi\right) \qquad (10)$$

$$P_M^\parallel(z) = \frac{P_0\eta_3A_rL}{z^2}(T_{mol}\beta_{mol}^\parallel + T_{aer}\beta_{aer}^\parallel) \times \exp\left(-2\int_0^z(\alpha_{mol}+\alpha_{aer})d\xi\right) \qquad (11)$$

where $P_C^\perp(z)$, $P_C^\parallel(z)$, $P_M^\parallel(z)$ are the backscattering signal of the perpendicular polarization channel, the parallel polarization channel, and the HSRL channel, respectively. $\beta_{mol}$ and $\beta_{aer}$ are the backscattering coefficients of molecular and aerosol, $\alpha_{mol}$ and $\alpha_{aer}$ are the extinction coefficients of molecular and aerosol. $P_0$ is the laser emitting power. $A_r$ is the telescope receiving area. $\eta_{1,2,3}$ is the optical efficiency of the receiving optics in each channel, and L is half of the pulse spatial transfer length.





$T_{mol}$ and $T_{aer}$ are the molecular and aerosol transmittances of the HSRL iodine filter, respectively, and z is the detection

distance.

By combining three equations, the extinction coefficient and backscattering coefficient of atmospheric aerosols can be obtained:

$$\beta_{aer}(z) = \frac{P_M(z) + P_C(z) \cdot T_{mol}}{P_M(z) - P_C(z) \cdot T_{aer}} \beta_{mol}(z) \tag{12}$$

$$\alpha_{aer}(z) = -\frac{1}{2} \frac{\partial}{\partial z} \left\{ \ln \left[ \frac{P_M(z)}{\beta_{mol}(z) T_{mol}} \right] \right\} - \alpha_{mol}(z) \tag{13}$$

**2.4.5 Relative difference between ACDL and ground-based lidar measurement**

We use the following relative error relation to assess the difference between two different instruments for the TABC and VDR

used in the comparison.

$$\text{bias}(z) = \frac{\text{ACDL }(z) - \text{G.B }(z)}{\text{G.B }(z)} \times 100 \tag{14}$$

where ACDL (z) and G.B (z) are the results of ACDL and ground-based lidar, respectively.

**3 Results and discussion**

Herein, we compare six individual cases of space-borne ACDL that overpassed the ground-based lidar sites, which are

classified into three weather conditions, including clear sky, dust events, and cloudy conditions. The ground-based lidar sites

used for comparison in this study are mainly the Zhangye site (38.9 °N, 100.5 °E, 1454 m) and Dunhuang site (40.1 °N, 94.6 °E,

1142 m) in BR-lidarnet. For the validation purpose, clear sky cases and cloudy cases in the daytime, and two dust cases in the

nighttime were selected. The specific comparison and processing process of data is shown in Figure 2.

**3.1 Clear Day Cases**

The ACDL overpasses the Zhangye site in the BR-lidarnet on June 10 and July 24, 2022, respectively. From Figures 3 and 4,

we can observe the presence of low TABC values below 5 km altitude in both cases and the VDR value is near 0.1. This

indicates that there are nearly spherical particles. Meanwhile, the air quality index (AQI) values of these two days in order are

43 and 45 (where the range of 0-50 is considered as no pollution), which exclude the possibility of pollution and therefore are

considered as clear sky days.

In order to further quantify and visually compare the observed discrepancies between ACDL and ground-based lidar, and to

eliminate the uncertainties such as random noise generated by the observations, 21 profiles were averaged in ACDL (i.e., take

the profile at the closest distance to the ground site is taken as the centre, and ten profiles are averaged before and after) and 5

profiles were averaged in the ground-based lidar (i.e., the transit time was cantered and averaged by taking a 10-minute profile

before and after each).

In terms of TABC in Figure 5, both ACDL and ground-based lidar observed the same vertical distribution characteristics, such

as the weaker aerosol layer below 5 km on June 10. In the altitude range of 2-5 km on June 10, the maximum values of TABC



measured by ACDL and ground-based lidar were 0.0014 and 0.0016 km$^{-1}$sr$^{-1}$. The observation results of the two are basically consistent. On July 24, the maximum values in the altitude range of 2-6 km were 0.0013 km$^{-1}$sr$^{-1}$ and 0.0015 km$^{-1}$sr$^{-1}$,

respectively. The observation results of the two are almost equal. It indicates that for TABC observation under clear weather conditions, the results obtained from ACDL and ground-based lidar observations are consistent, and the accuracy of ACDL observation is verified.

The signal-to-noise ratio of the ACDL is lower because the observation time was set as daytime, and the observation results are disturbed by the ambient signal noise, such as sunlight. The VDR values (Figure 5) of 4-5 km observed by the two on June

10 are consistent, with ACDL and ground-based lidar observing values of 0.097 and 0.088, respectively. Indicates the presence of spherical particles in this layer. In the case of July 24, the average VDR of ACDL and ground-based lidar below 5 km was 0.08 and 0.07, respectively.

**3.2 Dust Cases**

The ACDL overpassed the Dunhuang and Zhangye sites in BR-lidarnet on June 29 and July 10, 2022, respectively. Figure 6

illustrates that both the ACDL and ground-based lidar could detect the presence of a thick aerosol layer below 6 km with TABC values of 0.0017 km$^{-1}$sr$^{-1}$ and 0.0031 km$^{-1}$sr$^{-1}$, respectively. The VDR values reached 0.14 and 0.13, respectively, which indicate the existence of non-spherical particles, and therefore considered as a dust layer. According to the ground-based lidar observation, the dust layer lasted for about 2 hours.

The results in Figure 7 revealed that on July 10 over the Zhangye site, both ACDL and ground-based lidar could detect the

225 presence of a three-layer aerosol structure, but the third layer of aerosol observed by ground-based lidar is relatively thin, corresponding to altitudes of 2-3 km, 4-6 km, and 7-8 km above sea level, respectively. The observed and retrieved TABC values based on the ACDL and ground-based lidar in order were 0.0017 km$^{-1}$sr$^{-1}$ and 0.0023 km$^{-1}$sr$^{-1}$ in the altitude interval of 2-3 km, 0.0021 km$^{-1}$sr$^{-1}$ and 0.002 km$^{-1}$sr$^{-1}$ in the range of 4-6 km, and 0.0019 km$^{-1}$sr$^{-1}$ and 0.0017 km$^{-1}$sr$^{-1}$ in the range of 7-8 km. Similarly, the VDR values for the ACDL and ground-based lidar observations are 0.21 and 0.18 (for the altitude range of

230 2-3 km), 0.21 and 0.17 (for the altitude of 4-6 km), and 0.2 and 0.16 (for the range of 7-8 km), respectively. The observed VDR values represent the non-spherical particles and therefore could be judged as dust layer. The dust stratification is more noticeable as seen from the VDR values of the ground-based lidar.

Meanwhile, on July 10, there was a transit of the CALIPSO satellite. The shortest distance between CALIPSO and ACDL observation trajectories is 68.65 km, with a time difference of 80 minutes between passing through ground stations. The

235 CALIPSO observation is consistent with ACDL, and three layers of dust layers are observed (see Figure 7 right column).

By further quantifying and comparing the discrepancies between the results of ACDL and those of the ground-based lidar, the observations of the ACDL and those of the ground-based lidar are in better agreement in the presence of dust aerosol (see Figure 8). In Dunhuang, a dust layer below 6 km was observed by both ACDL and ground-based lidar (see Figure 8 left). The comparison shows that the TABC values below 4 km based on the ground-based lidar are larger (0.0036 km$^{-1}$sr$^{-1}$) than ACDL

(0.0021 km$^{-1}$sr$^{-1}$), probably due to the inability of the ACDL laser to penetrate the thick dust layer when observed from above,





but above 4 km the ACDL correspond better. The comparison of observations at the Zhangye site is more consistent, and the three dust layers correspond well to each other in height (see Figure 8 right). The bias of the TABC on dusty days is -10.5 ± 25.4%.

According to the VDR observation values, the aerosol layer positions are all between 0.2-0.3 as dust aerosols. The same dust
layer positions are observed by both the ACDL and ground-based lidar. The comparison of VDR values at the two sites are both consistent, and the bias of VDR on dust day was -6.0 ± 38.5%. However, the signal-to-noise ratio of ACDL observation results is slightly lower than ground-based lidar.

Comparing the observation results of CALIPSO and ACDL (see Figure 9), it was found that the two observations were almost identical and had a good consistency. Both in terms of observed values and the structure of the dust layer, the ACDL
observation results show good consistency with the CALISO observation results.

**3.3 Cloudy Day Cases**

The ACDL overpassed Zhangye and Dunhuang sites in BR-lidarnet on May 27 and July 6, 2022, respectively. The ACDL observation on the left in Figure 10 reveals that the clouds spread over a large area with the horizontal extent of the clouds greater than 200 km. The observed cloud base and cloud top heights are 5 km and 10 km, respectively. The historical weather
forecast also shows a cloudy condition over Zhangye on that day. According to the VDR, the value is around 0.4, which is considered as a cirrus cloud. The cloud is identified as stratiform based on the spatial distribution characteristics observed by the ACDL. The low signal-to-noise ratio below 6 km is probably due to the fact that the ACDL laser signal does not penetrate deeply into the cloud and the signal is scattered and attenuated several times within the cloud. Based on the ground-based lidar observation on the right side of Figure 10, the cloud is very long-lived and has existed during the observation period. The
height of the cloud base and top in order are 4 km and 7 km, which are lower than the height observed by the ACDL. The VDR values indicate the presence of ice crystals within the cloud, which is a cirrus cloud. The signal-to-noise ratio is reduced above 5 km as seen from the VDR signal, probably because the cloud layer is too thick and the laser from the ground-based lidar cannot be effectively penetrated, and the signal is scattered and attenuated several times inside the cloud, which also leads to the reduced signal-to-noise ratio.

On July 6, Dunhuang was also recorded as a cloudy day as observed from historical weather. Thick clouds in the altitude range of 8-10 km were observed by the ACDL (see Figure 11), but TABC from ground-based lidar only showed a thin high-value area at 7.5 km. Based on the VDR of ground-based lidar, it was observed that the signal-to-noise ratio is reduced above 7 km, probably due to the presence of thick clouds, which causes the lidar laser not to penetrate the clouds, thus resulting in a bias between the inversion and the observation. The VDR value observed by both the ACDL and ground-based lidar was 0.3, which
probably depicted a mixed-phase cloud composed of ice crystals and supercooled liquid droplets.

Through further quantifying to compare the discrepancy in observations between the ACDL and the ground-based lidar, the height of clouds obtained by the ground-based lidar is lower than that of the ACDL (see Figure 12). However, numerically both TABC and VDR values are closer. It is also apparent that for the ACDL observations, the signal-to-noise ratio in the





presence of the clouds is lower, so the profile is not smooth. Based on the ground-based lidar observations, the signal-to-noise
ratio above the cloud is lower. This issue is mainly attributed to the attenuation of the signal by the clouds.

Since the presence of clouds affects the observations and retrievals, therefore, we focus on four distinct cases under cloud-free
conditions (see Figure 13). In general, these two observations are in better agreement. In particular, the results of the VDR are
better with an observed bias of -8.5 ± 35.8% and $R^2$ of 0.91. The observation results used for comparison are all within the EE
lines. The overall data sample of the TABC is closer to the 1:1 line, with an observed bias of -9.8 ± 19.1% and $R^2$ of 0.65.
Due to the Raman channels can only be detected at night, the dust case of ACDL overpassing the Dunhuang site on June 29
was selected to compare the 532 nm extinction coefficient and lidar ratio obtained by ground-based lidar Raman retrieval
method and ACDL HSRL retrieval method (two panels below Figure 13). The extinction coefficients obtained by the two are
relatively consistent, with a bias of 1.23 ± 53.62%. The bias of the lidar ratio is -22.75 ± 30.37%.

## 4 Conclusions

In this work, initial observations from the first high resolution space-borne lidar were validated against the ground-based lidar
network. The ACDL retrieved profiles of TABC and VDR of level-2A products are compared with the coincident lidar
measurements for the six cases under various weather conditions (See Table 1 for a detailed summary). We first convert the
detection direction of the ground-based lidar to the same direction as the ACDL and then compare the observation and retrieval
results of these lidars under three weather conditions (i.e., clear sky, dust, and cloudy).

For clear days, both lidars observed the same vertical distribution characteristics. Since both individual observations for clear
days are in the daytime, therefore, the overall signal-to-noise ratio is lower due to the influence of the ambient noise. On the
other hand, the observed results of the VDR from both lidars are relatively consistent.

In the presence of dust aerosols, the retrieval results of the ACDL and ground-based lidar match better and both could observe
the same structure of the dust layer. In the case of a thicker dust layer, it could cause the ACDL to fail to penetrate, so there
would be some deviation between the two results near the ground. Since both individual observations are conducted at
nighttime, the signal-to-noise ratio is high and the observations are in good agreement. The calculated biases of the TABC and
VDR in order are -10.5 ± 25.4 % and -6.0 ± 38.5 %, respectively. In addition, corresponding to the dust case on July 10, there
was an overpass of the CALIPSO satellite, and it was found that the observation results of ACDL and CALIPSO were highly
consistent.

In the presence of a thicker layer of clouds, the comparison indicated that the cloud height obtained from the ground-based
lidar inversion is lower than that observed by the ACDL due to the presence of thick clouds that block the laser transmission.
In the absence of clouds, both lidars provide better results for the VDR with an observed deviation of -8.5 ± 35.8% and $R^2$ of
0.91. The observed bias of the TABC values is -9.8 ± 19.1% and $R^2$ is 0.65.

Finally, one of the detection advantages of ACDL is high spectral lidar detection, which can retrieve the extinction coefficient and the lidar ratio profile of aerosols without assuming lidar ratio. In order to compare the accuracy of its high spectral lidar detection, the extinction coefficient obtained from the Raman channel retrieval of ground-based lidar was compared with the lidar ratio and ACDL results. Due to the Raman channels can only be detected at night, and combined with the comparison results of ACDL and ground-based lidar in TABC and VDR, the dust case on June 29 was selected for validation. The

comparison results showed that for the case of ACDL passing through Dunhuang, the extinction coefficient profiles obtained by the two were relatively close, with a total bias of $1.23 \pm 53.62$ %. The bias of the lidar ratio is $-22.75 \pm 30.37$ %.

The present exploration proves that the ACDL provides a reliable observation of aerosols and clouds subjected to various conditions, which helps to further evaluate the effects of aerosol on climate and environment as well as the ecosystem in the future. However, there are still some discrepancies between the two observations, and more ground-based lidar observation

should be employed for the validation of ACDL on a global scale in the future.

**Author contribution**

Qiantao Liu performed the analysis and wrote the paper. Jiqiao Liu, Guangyao Dai and Xiaodong Song performed the original data. Qingqing Dong and Meishi Li were helpful with the methods used in the article. Wuren Li and Ze Li provided technical support. Qiantao Liu and Zhongwei Huang interpreted the results. Zhongwei Huang and Jiqiao Liu provided significant input

to the interpretation and the improvement of the paper. All authors discussed the results and commented on the paper.

**Competing interests**

The authors declare that they have no conflict of interest.

**Data availability**

The ACDL data we used in this paper are not available publicly at the time when the article was submitted. We are allowed to

access the data through our participation as a part of ACDL scientific team. The CALIPSO lidar observation data are freely available from Atmospheric Science Data Center (NSDC) website (https://subset.larc.nasa.gov/calipso/).

**Acknowledgements**

This work was supported in part by the Second Tibetan Plateau Scientific Expedition and Research Program (STEP), Grant No. 2019QZKK0602; Gansu Provincial Science and Technology Innovative Talent Program: High-level Talent and Innovative

Team Special Project (No.22JR9KA001); Fundamental Research Funds for the Central Universities (lzujbky-2022-kb10 and lzujbky-2022-kb11). The authors would like to thank the operators for providing ACDL level-2 data and HSRL channel data.



The data of ground-based lidar in BR-lidarnet provided by the corresponding author. The maps in figures were exported using Goggle Earth (version: 10.38.0.0, Apache License Version 2.0, January 2004 https://www.apache.org/licenses/), we use it on the website (https://google.cn/earth/) and we have adhered to their redistribution permissions.

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

**Figure captions**

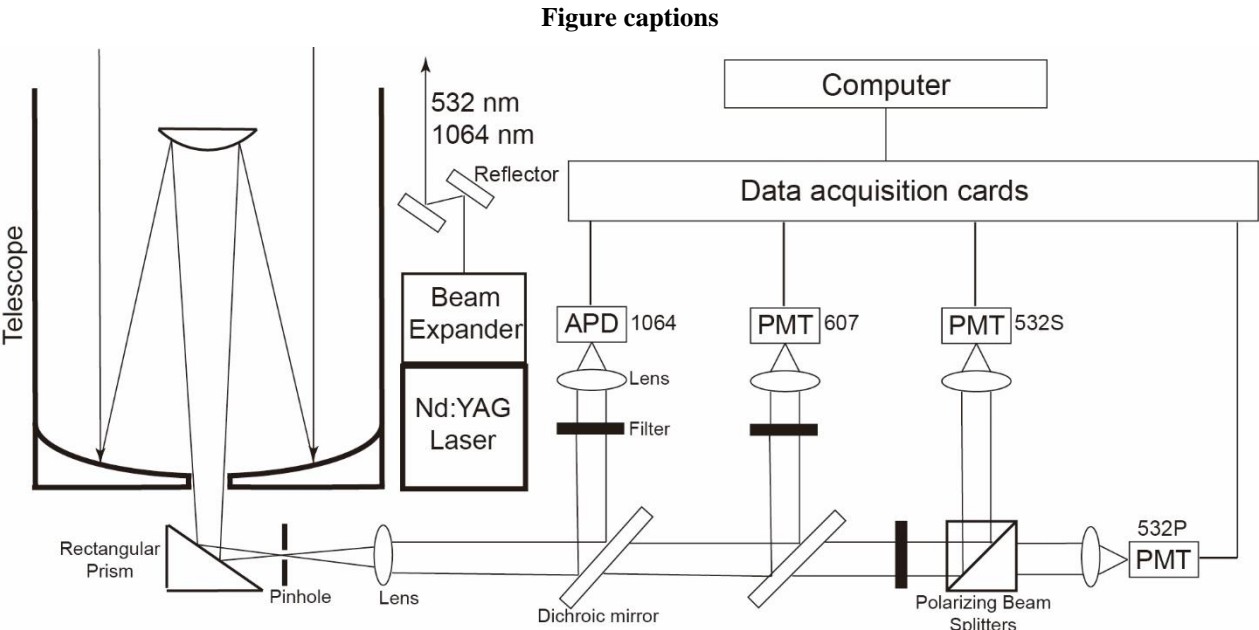

**Figure 1: Schematic diagram of the developed ground-based lidar system of the BR-lidarnet employed in the present investigation.**







Figure 2: Sketch of the comparison between ACDL and Ground-based Lidar in the different weather conditions.



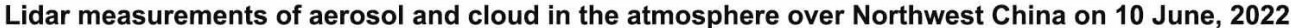

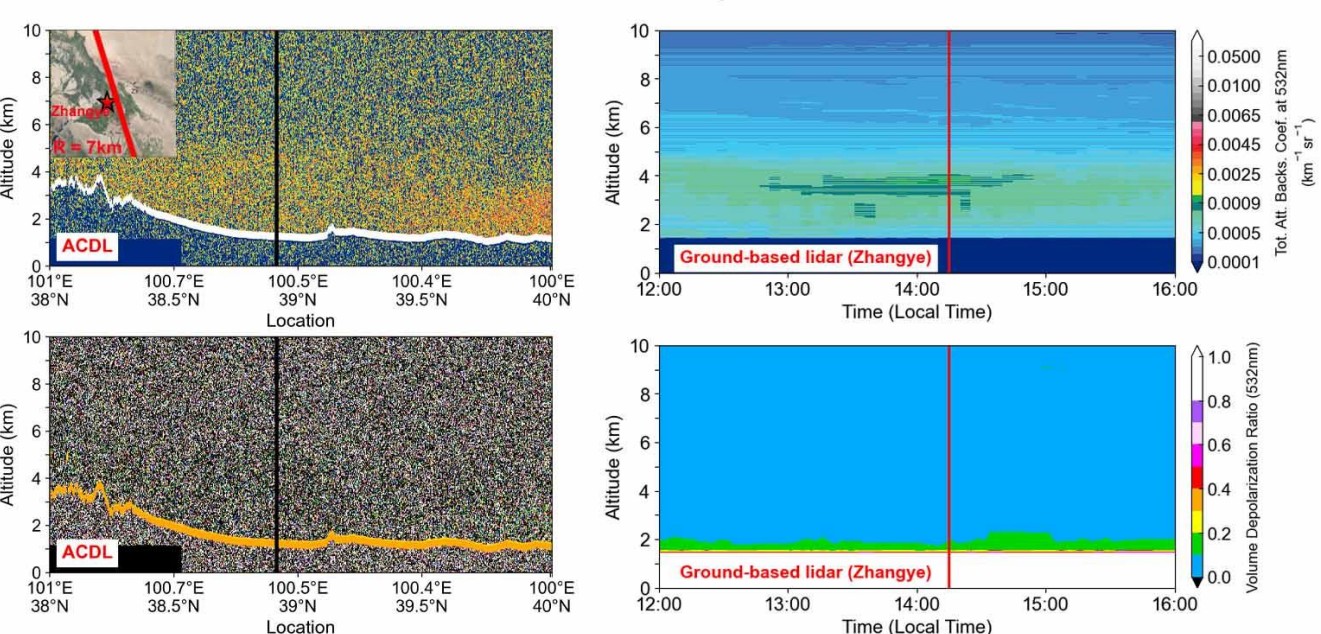

**Figure 3: Vertical structure of atmospheric aerosols and clouds observed via the ACDL and ground-based lidar on 10 June 2022. The black solid line signifies the closest point (7 km) to the Zhangye site in the ACDL overpass track. At the right panel, the red solid line corresponds to the ACDL overpass time (14:25 LT). The red pentagram in the upper-left panel indicates the location of the ground-based lidar site and the ACDL track.**

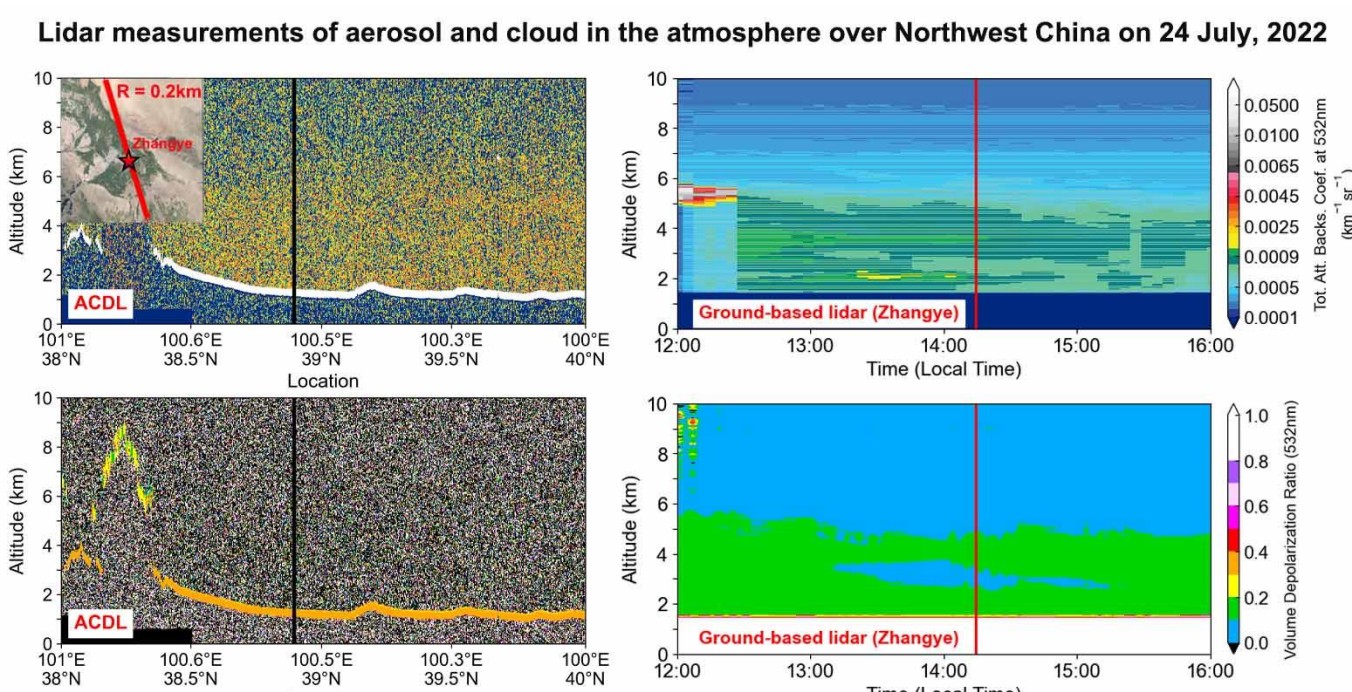

**Figure 4: Vertical structure of atmospheric aerosols and clouds observed via the ACDL and ground-based lidar on a clear day on 24 July 2022. The closest distance between the ACDL track and the ground-based lidar is 0.2 km. The ACDL overpass time is 14:24 LT.**







**Figure 5: Comparison between the TABC and VDR profiles at 532 nm measured by the ACDL (black solid line) and ground-based lidar (lime solid line) on 10 June 2022 and 24 July 2022. The shaded envelope represents the standard deviation.**



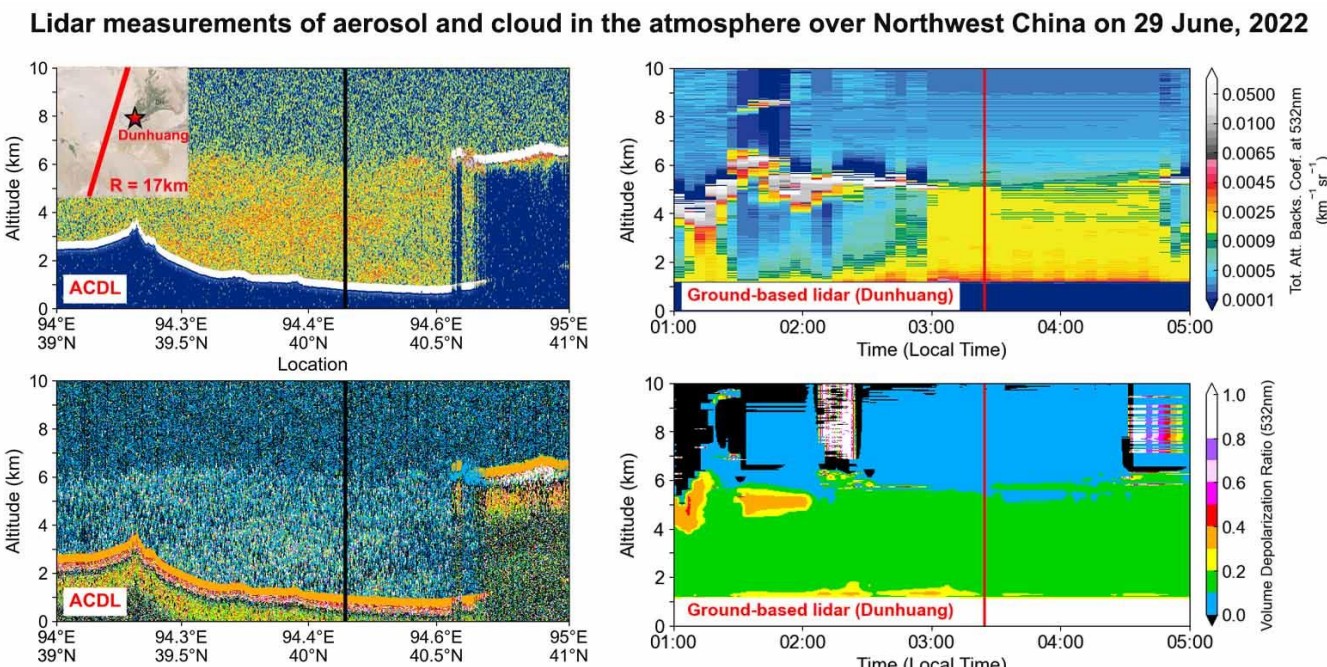

**Figure 6: Vertical structure of atmospheric aerosols and clouds observed via the ACDL and ground-based lidar for a fairly dusty day on June 29, 2022. The closest distance between the ACDL track and the Dunhuang lidar site was 17 km. The ACDL overpass time is 3:41 LT.**

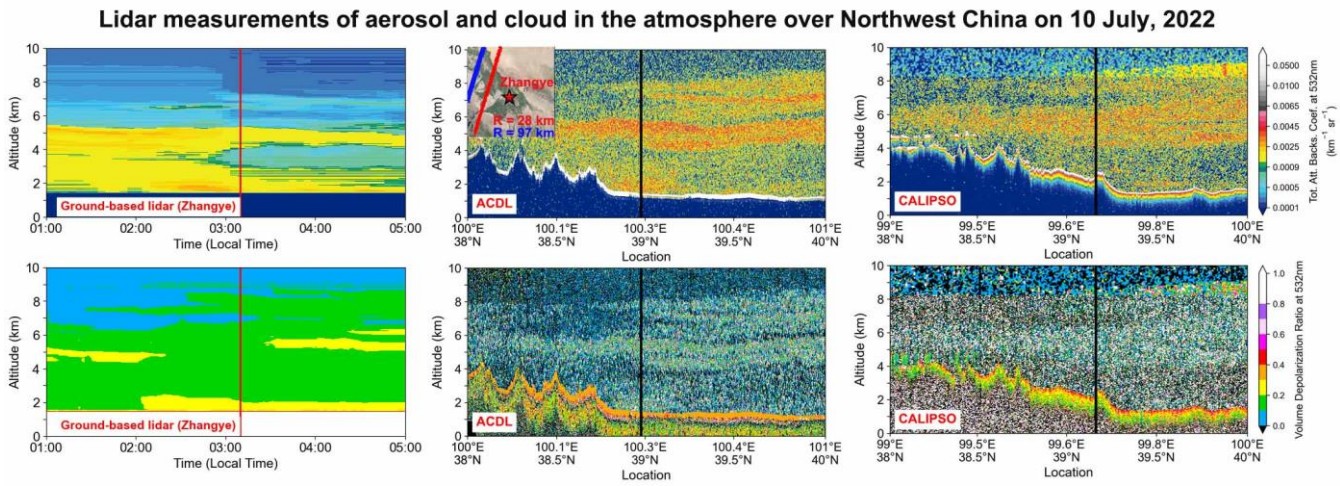

**Figure 7: Vertical structure of atmospheric aerosols and clouds observed via the ground-based lidar (left), ACDL (centre), and CALIPSO (right) on July 10, 2022. The closest distance between the ACDL track (red line) and the Zhangye lidar site is 28 km. The ACDL overpass time is 3:16 LT. The closest distance between the ACDL track and the CALIPSO track (blue line) is 68.65 km.**




**Figure 8: Comparison between the TABC and VDR profiles at 532 nm measured by the ACDL and ground-based lidar on 29 June 2022 and 10 July 2022 respectively. The shaded envelope represents the standard deviation.**



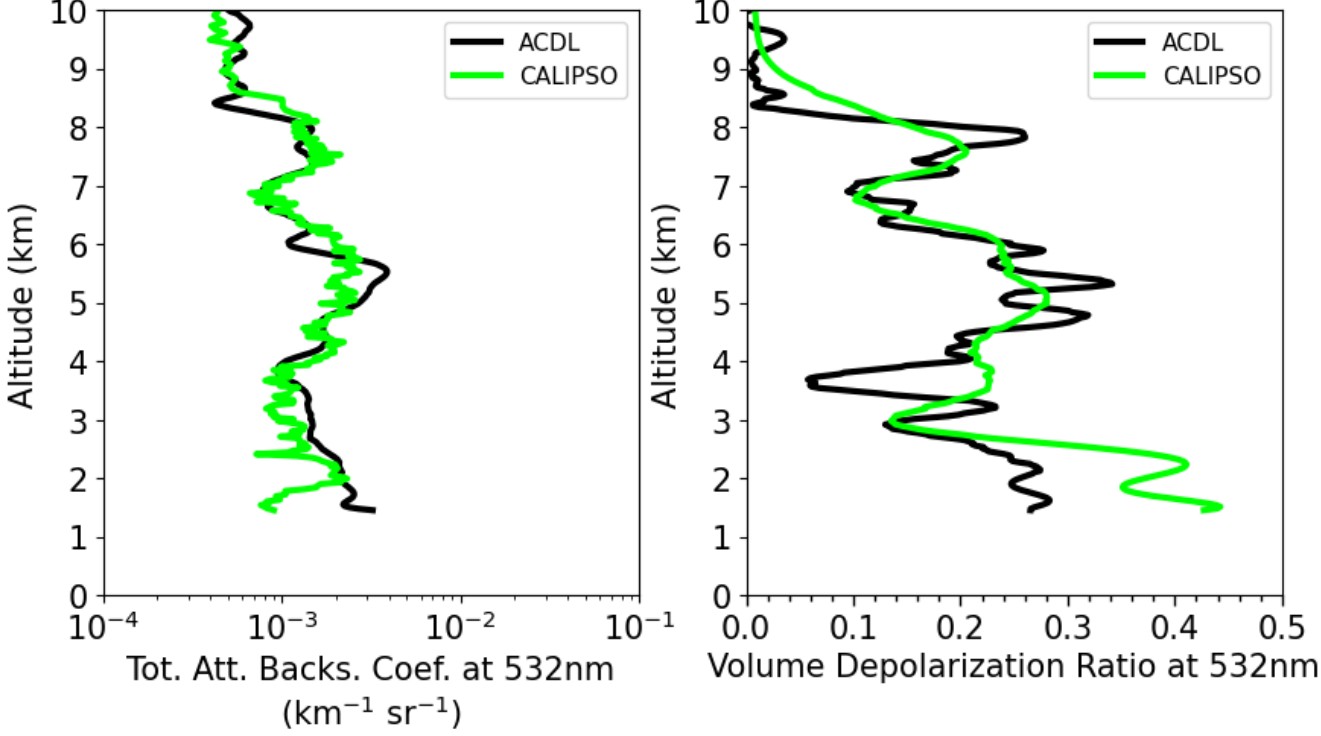

**Figure 9: Comparison between the TABC and VDR profiles at 532 nm measured by the ACDL (black solid line) and CALIPSO (lime solid line) on 10 July 2022.**







**Figure 10: Vertical structure of atmospheric aerosols and cloud observed via the ACDL and ground-based lidar for a cloudy day on May 27, 2022 (i.e., the closest distance between the ACDL track and Zhangye lidar site is 13 km and the overpass time of ACDL is 3:17 LT).**



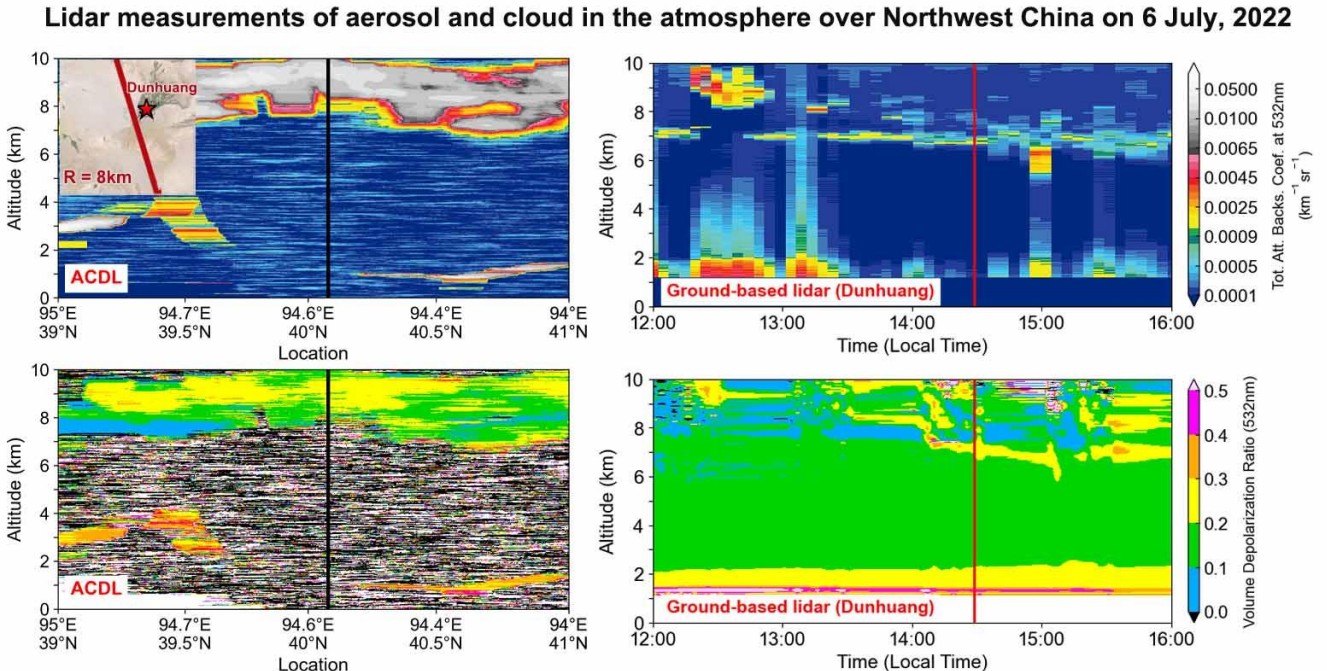

**Figure 11: Vertical structure of atmospheric aerosols and clouds observed via the ACDL and ground-based lidar on July 6, 2022. The closest distance between the ACDL track and the Dunhuang lidar site is 8 km. The ACDL overpass time is 14:48 LT.**







**Figure 12: Comparison between the TABC and VDR profiles at 532 nm measured by the ACDL (black solid line) and ground-based**
**lidar (lime solid line) on 27 May 2022 and 6 July 2022 respectively. The shaded envelope represents the standard deviation.**



## Comparison between ACDL and Ground-based lidar observations

**Figure 13: Scatterplots of the ground-based lidar retrieved TABC (upper left) and VDR (upper right) versus ACDL measurement under cloud-free conditions. The solid line refers to the 1:1 line; the dashed line refers to the expected error envelope; EE = ± (0.0005 + 0.3 × TABC$_{Ground}$); EE = ± (0.05 + 0.3 × VDR$_{Ground}$). The two panels below the figure show the comparison between the extinction coefficient and lidar ratio obtained from ACDL and ground-based lidar retrieval under the dust case on June 29th. The solid line refers to the 1:1 line; the dashed line refers to the expected error envelope; EE = ± (0.5x10$^{-4}$ + 0.3 × Ext$_{Ground}$); EE = ± (10 + 0.3 × LR$_{Ground}$).**





| Observed Time (Local time) | Overpassed Site | Closest distance point | Closest distance (km) | Weather Condition | TABC Bias (%) | VDR Bias (%) |
|---|---|---|---|---|---|---|
| 2022.6.10 14:25 | Zhangye | 100.5°E, 38.9°N | 7.0 | Clear | -3.93±13.62 | -16.28±35.79 |
| 2022.7.24 14:24 | Zhangye | 100.5°E, 38.9°N | 0.2 | Clear | -14.49±12.04 | -5.73±30.49 |
| 2022.6.29 03:41 | Dunhuang | 94.6°E, 40.1°N | 17.3 | Dust | -25.15±18.84 | -18.24±40.79 |
| 2022.7.10 03:16 | Zhangye | 100.5°E, 38.9°N | 28.6 | Dust | 4.25±32.01 | 6.27±36.23 |
| 2022.5.27 03:17 | Zhangye | 100.5°E, 38.9°N | 13.0 | Cloudy | -8.33±30.53 | -9.07±54.96 |
| 2022.7.6 14:48 | Dunhuang | 94.6°E, 40.1°N | 8.4 | Cloudy | -4.68±23.59 | -6.12±41.57 |

**Table 1: Summary of ACDL overpassed the ground-based lidar sites.**