# Peer review of "Validation of initial observation from the first space-borne high spectral resolution lidar with ground-based lidar network"

_Atmospheric Measurement Techniques, 2023_

## Author Comment (AC1)

**Reply to the comments of Reviewers**

**RC1: 'Comment on amt-2023-235', Anonymous Referee #1:**

This study reports initial validation of the world's first space borne HSRL lidar, launched by China. Results show overall high consistency with ground-based Raman lidar and CALIPSO satellite measurements, which proves the reliability of the HSRL lidar technique in retrieving aerosol profiles in space. The findings are valuable and the study is timely. However, I have a few concerns, mainly about data availability, and feel some improvements are needed.

**Response:** We would like to thank the reviewer for providing constructive suggestions and comments. We have revised the manuscript based on the reviewer's comments by providing the following point-by-point responses.

Specific comments:

1. In the acknowledgement the authors indicated that the space HSRL data is not publicly available at the time of submission. I wonder if the data is available now? This is important since the results of the paper cannot be verified or repeated if the data cannot be accessed.

**Response:** Thanks for your question. Data are publicly available, although they not yet released to the public, but have been shared with relevant teams for retrieval algorithms and validation. Some research articles have already been published by these teams (Liu et al., 2019; Ke et al., 2022; Dai et al., 2023; Hu et al., 2023; Zha et al., 2023; Wang et al., 2023). Data from our manuscript may be politely requested from the corresponding author if the readers need it.

Reference:

Dai, G., Wu, S., Long, W., Liu, J., Xie, Y., Sun, K., Meng, F., Song, X., Huang, Z., and Chen, W.: Aerosols and Clouds data processing and optical properties retrieval algorithms for the spaceborne ACDL/DQ-1, Aerosols/Remote Sensing/Data Processing and Information Retrieval, https://doi.org/10.5194/egusphere-2023-2182, 2023.

Hu, J., Wang, X., Zhao, S., Wang, Z., Yang, J., Dai, G., Xie, Y., Zhu, X., Liu, D., Hou, X., Liu, J., and Chen, W.: Spaceborne High Spectral Resolution Lidar for Atmospheric Aerosols and Clouds Profiles Measurement, Acta Optica Sinica, 43, 18, 10.3788/AOS231153, 2023.

Ke, J., Sun, Y., Dong, C., Zhang, X., Wang, Z., Lyu, L., Zhu, W., Ansmann, A., Su, L., Bu, L., Xiao, D., Wang, S., Chen, S., Liu, J., Chen, W., and Liu, D.: Development of China's first space-borne aerosol-cloud high-spectral-resolution lidar: retrieval algorithm and airborne demonstration, PhotoniX, 3, 17, https://doi.org/10.1186/s43074-022-00063-3, 2022.

Liu, D., Zheng, Z., Chen, W., Wang, Z., Li, W., Ke, J., Zhang, Y., Chen, S., Cheng, C., and Wang, S.: Performance estimation of space-borne high-spectral-resolution lidar for cloud and aerosol optical properties at 532 nm, Opt. Express, 27, A481, https://doi.org/10.1364/OE.27.00A481, 2019.

Wang, J., Liu, D.: Comparison and Analysis of Payloads Performance for Active and Passive Spaceborne Atmospheric Detection, Acta Optica Sinica, 43, 18, 10.3788/AOS231153, 2023.

Zha, C., Bu, L., Li, Z., Wang, Q., Mubarak, A., Liyanage, P., Liu, J., and Chen, W.: Aerosol Optical Properties Measurement using the Orbiting High Spectral Resolution Lidar onboard DQ-1 Satellite: Retrieval and Validation, Atmos. Meas. Tech. Discuss. [preprint], https://doi.org/10.5194/amt-2023-219, in review, 2023.

2. Although reasonable agreement is reached between HSRL and Raman lidar results, remarkable differences still show up in some cases and in the scatter plots such as Figure 13, especially for the depolarization ratio. It is therefore necessary to discuss the sources of uncertainties or causes of the differences. However, I did not seem to notice these discussions in the paper;

**Response:** Thanks for your suggestion. The discrepancies in the comparison of aerosol and cloud between ACDL and ground-based lidar measurement are possible because of the several factors such as different lidar system parameters, the difference in detection distance between spaceborne and ground-based systems, various aerosols

and clouds distribution, the inhomogeneity of the atmosphere and also can be due to the uneven terrain in the northwest region of China. Furthermore, even though the measurements are simultaneous, if the observation points are not at the same location, it can lead to discrepant comparison results (Chiang et al., 2011).

Typically, the main source of uncertainty in the depolarization ratio is the systematic errors in the optical setup of the lidar systems (Belegante et al., 2018). In particular, different field of view (FOV) can lead to different multiple scattering effects. The FOV of ACDL is 0.2 mrad, the pulse energy is 150 mJ and telescope aperture is Φ1000 mm (Dong et al., 2018; Liu et al., 2019). The FOV of ground-based lidar is 0.5 mrad, the pulse energy is 100 mJ and telescope aperture is Φ200 mm. The experiment found that by changing the field of view angle, the lidar and the aerosol remain unchanged, resulting in different depolarization ratios. This is due to the multiple scattering effects, which are more pronounced in the clouds (Hu et al., 2006).

The random noise in the ACDL profiles is much higher than the ground-based lidar profiles due to their much longer measurement range and shorter integration times, and the daytime noise level is higher than nighttime due to the statistical fluctuations associated with solar background light. Reducing the noise to the required level requires averaging profiles along the ground track (Gimmestad et al., 2017). However, spatial aerosol inhomogeneities introduce systematic error into the averages. The Zhangye and Dunhuang lidar station is located at 1454 m a.s.l. and 1142 m a.s.l. with a complex topography of the surrounding area (Figure R1) that makes very difficult the comparison in the planetary boundary layer with satellite data acquired with imperfect spatial coincidence. Overall, the observation comparison results between the ACDL and ground-based lidar are acceptable.

In the end of Section 3, we added "***The discrepancies in the comparison between ACDL and ground-based lidar measurement are possible because of the several factors such as different lidar system parameters, the difference in detection distance between spaceborne and ground-based systems and also can be due to the inhomogeneity of the atmosphere (Chiang et al., 2011; Belegante et al., 2018). Overall, the observation comparison results between the ACDL and ground-based***

*lidar are acceptable.*" in lines 285-290. And we have changed the distance details in Figure 2.

[Figure]

**Figure R1.** Topographic characteristics of the location of ground-based lidar.

Reference:

Belegante, L., Bravo-Aranda, J. A., Freudenthaler, V., Nicolae, D., Nemuc, A., Ene, D., Alados-Arboledas, L., Amodeo, A., Pappalardo, G., D'Amico, G., Amato, F., Engelmann, R., Baars, H., Wandinger, U., Papayannis, A., Kokkalis, P., and Pereira, S. N.: Experimental techniques for the calibration of lidar depolarization channels in EARLINET, Atmos. Meas. Tech., 11, 1119–1141, https://doi.org/10.5194/amt-11-1119-2018, 2018.

Chiang, C.-W., Kumar Das, S., Shih, Y.-F., Liao, H.-S., and Nee, J.-B.: Comparison of CALIPSO and ground-based lidar profiles over Chung-Li, Taiwan, Journal of Quantitative Spectroscopy and Radiative Transfer, 112, 197–203, https://doi.org/10.1016/j.jqsrt.2010.05.002, 2011.

Dong, J., Liu, J., Bi, D., Ma, X., Zhu, X., Zhu, X., and Chen, W.: Optimal iodine absorption line applied for spaceborne high spectral resolution lidar, Appl. Opt., 57, 5413, https://doi.org/10.1364/AO.57.005413, 2018.

Hu, Y., Liu, Z., Winker, D., Vaughan, M., Noel, V., Bissonnette, L., Roy, G., and McGill, M.: Simple relation between lidar multiple scattering and depolarization for water clouds, Opt. Lett., 31, 1809, https://doi.org/10.1364/OL.31.001809, 2006.

Gimmestad, G., Forrister, H., Grigas, T., and O'Dowd, C.: Comparisons of aerosol backscatter using satellite and ground lidars: implications for calibrating and validating spaceborne lidar, Sci Rep, 7, 42337, https://doi.org/10.1038/srep42337, 2017.

Liu, D., Zheng, Z., Chen, W., Wang, Z., Li, W., Ke, J., Zhang, Y., Chen, S., Cheng, C., and Wang, S.: Performance estimation of space-borne high-spectral-resolution lidar for cloud and aerosol optical properties at 532 nm, Opt. Express, 27, A481, https://doi.org/10.1364/OE.27.00A481, 2019.

3. It is not clear how the space lidar and surface lidar are collocated? And how space HSRL and CALIPSO are collocated? Lidar orbit tracks are typically narrow and there may be certain distances between the lidar orbit and the surface lidar. Also, it is not clear how lidar signals are averaged spatially (vertical and horizontal) and temporally? The ground lidar data presented seem to have lower vertical resolution so it seems some averaging is performed.

**Response:** We thank you for raising the question. In order to better illustrate the collocating of ACDL and ground-based lidar, we have made a schematic diagram (Figure R2). Collocating is centered around the location of the ground-based lidar (red star in the Figure R2), with a radius of 50 km drawn as a circle (the circle represented by the black dashed line in the Figure R2). The observation trajectory of ACDL is very narrow, so we can approximate it as a line (green line in the Figure R2). If this line intersects with the drawn circle, we believe that the observation results of the two could be compared. The orange dashed line in the Figure R2 represents the observation distance between the ACDL and ground-based lidar. The trajectory matching method of CALIPSO is the same as ACDL, and the collocate between CALIPSO and ACDL is derived from their distances from ground-based lidar. In the case of July 10, 2022, the nearest position of CALIPSO ground track is 97 km away from Zhangye ground-based lidar station and 69 km away from ACDL, which we think is a comparable range.

[Figure]

**Figure R2.** Schematic diagram of position matching and data selection for spaceborne and ground-based lidar.

We use moving average to smooth the vertical signals of both spaceborne and ground-based lidars. To reduce the random error of instrument observation, we use the location nearest the ground-based lidar position in the ground track of the spaceborne lidar (point O in Figure R2) as the center. We take 10 profiles before and after it (a total of 21 profiles) for averaging. The relevant schematic diagram is located in the upper right corner of Figure R2. For ground-based lidar, we take the satellite observation time corresponding to point O as the overpass time of the spaceborne lidar. Based on this time, we select 10 minutes of data before and after the overpass time and average them (the time resolution of the ground-based lidar is 5 minutes, so there is a total of 5 profiles). The relevant schematic diagram is located in the lower right corner of Figure R2. In Section 3, we added "***Selecting satellite ground tracks passing within a 50 km radius centred on ground-based lidar for validation.***". And we moved "***In order to further quantify and visually compare the observed discrepancies between ACDL and ground-based lidar, and to eliminate the uncertainties such as random noise generated by the observations, 21 profiles were averaged in ACDL (i.e., take the***

*profile at the closest distance to the ground site is taken as the centre, and ten profiles are averaged before and after) and 5 profiles were averaged in the ground-based lidar (i.e., the overpass time was cantered and averaged by taking a 10-minute profile before and after each).*" in Section 3.1 to Section 3.

4. The HSRL was launched more than 1.5 years ago, which should have obtained quite large amounts of data. So why only 6 cases are selected? How are they selected?

**Response:** Thanks for your suggestion. The focus of our work is to evaluate the observation performance of ACDL in the first few months, as stated in the title of the manuscript as "Initial". Because the quality of data in these months is extremely important for evaluating the observational performance of new launched spaceborne remote sensing instruments. After the instrument starts working, it needs to take some time to reach a stable observation state. Therefore, we selected the data of the Belt and Road lidar network from May to July to match the six best-matched cases in time and space. And these 6 cases include 3 different typical types of clear, dust, and cloudy atmospheric scenes, with observation times also including daytime and nighttime.

The matching method is centered around the location of the ground-based lidar (red star in the Figure R2), with a radius of 50 km drawn as a circle (the circle represented by the black dashed line in the Figure R2). The ground track of ACDL is the green line in the Figure R2. If this line intersects with the drawn circle, we believe that the observation results of the two can be compared. We have changed the sentence in Section 3: "*Herein, we compare six best-matched cases of space-borne ACDL that overpassed the ground-based lidar sites in time and space, which are classified into three weather conditions, including clear sky, dust events, and cloudy conditions.*".

5. Why not also use EARLINET data? It is publicly available and may increase valid cases.

**Response:** Thanks for your suggestion. Validation is extremely important for a newly launched spaceborne lidar, especially in different regions, times, and weather conditions. Ground-based lidar data from different networks should be used to evaluate

the performance of ACDL. However, in order to evaluate the initial observation performance of ACDL as soon as possible, we firstly used the self-developed muti-wavelength Raman polarization lidar network data and CALIPSO data. In the future, we hope to use more data with other teams around the world (such as EARLINET, ADNET, MPLNET) to evaluate the observational performance of ACDL in other places.

6. Some technical basics of the HSRL, such as signal to noise ratio, calibration method, data quality control, should be described.

**Response:** Thanks for your suggestion. The manuscript only provides a brief description of HSRL technology. For more details, please refer to Dai et al. (2023) published article. We added "***For specific data processing methods such as signal-to-noise ratio processing, calibration method and data quality control, please refer to relevant published articles (Liu et al., 2019; Ke et al., 2022; Dai et al., 2023; Zha et al., 2023).***" in section 2.4.4 High spectral resolution lidar (HSRL) retrieval method from ACDL.

Reference:

Dai, G., Wu, S., Long, W., Liu, J., Xie, Y., Sun, K., Meng, F., Song, X., Huang, Z., and Chen, W.: Aerosols and Clouds data processing and optical properties retrieval algorithms for the spaceborne ACDL/DQ-1, Aerosols/Remote Sensing/Data Processing and Information Retrieval, https://doi.org/10.5194/egusphere-2023-2182, 2023.

Ke, J., Sun, Y., Dong, C., Zhang, X., Wang, Z., Lyu, L., Zhu, W., Ansmann, A., Su, L., Bu, L., Xiao, D., Wang, S., Chen, S., Liu, J., Chen, W., and Liu, D.: Development of China's first space-borne aerosol-cloud high-spectral-resolution lidar: retrieval algorithm and airborne demonstration, PhotoniX, 3, 17, https://doi.org/10.1186/s43074-022-00063-3, 2022.

Liu, D., Zheng, Z., Chen, W., Wang, Z., Li, W., Ke, J., Zhang, Y., Chen, S., Cheng, C., and Wang, S.: Performance estimation of space-borne high-spectral-resolution lidar for cloud and aerosol optical properties at 532 nm, Opt. Express, 27, A481, https://doi.org/10.1364/OE.27.00A481, 2019.

Zha, C., Bu, L., Li, Z., Wang, Q., Mubarak, A., Liyanage, P., Liu, J., and Chen, W.: Aerosol Optical Properties Measurement using the Orbiting High Spectral Resolution Lidar onboard DQ-1 Satellite: Retrieval and Validation, Atmos. Meas. Tech. Discuss. [preprint], https://doi.org/10.5194/amt-2023-219, in review, 2023.

---

## Author Comment (AC2)

**Response to the comments of Reviewers**

**RC2: 'Comment on amt-2023-235', Anonymous Referee #2**

The article introduces China's first successful launch of the world's first space hyperspectral resolution laser mine (ACDL), and analyzes the observation results of six cases including sunny, dust storm, and cloudy weather with ground-based LiDAR and CALIPSO, verifying that the radar can accurately detect the three-dimensional distribution of aerosols and clouds worldwide. However, the introduction and results analysis were not well described, resulting in redundant components and analysis deficiencies.

**Response:** We would like to thank the reviewer for providing constructive suggestions and comments. We have addressed each of the reviewer's comments and it has improved the quality and clarity of the manuscript.

Specific comments:

1. In the second and third paragraphs of the introduction, the author provides a large number of examples to prove the argument that satellite based LiDAR data must be appropriately validated by ground-based LiDAR. In my opinion, this is too redundant. Two typical examples are sufficient.

**Response:** Thanks for your suggestion. Indeed, the introduction section provides too much description of the satellite validation work. Based on the suggestions of the referee, we have streamlined the content of the introduction.

2. In Figures 5 and 8, the TABC results of ACDL generally show a left shift phenomenon compared to the results of ground-based LiDAR, and the fluctuation of ACDL is significantly larger. Can the author explain this phenomenon?

**Response:** We thank you for raising the question. The discrepancies in the comparison of aerosol and cloud between ACDL and ground-based lidar measurement are possible because of the several factors such as different lidar system parameters, the difference in detection distance between spaceborne and ground-based systems, various

aerosols and clouds distribution, the inhomogeneity of the atmosphere and also can be due to the uneven terrain in the northwest region of China (Figure R1). Furthermore, even though the measurements are simultaneous, if the observation points are not at the same location, it can lead to discrepant comparison results (Chiang et al., 2011). Due to the closer observation positions of the two clear cases in Figure 5, the observation difference between Figure 5 is smaller than that in Figure 8 (the closest distance in Figure 5 are 7 km and 0.2 km, respectively; the closest distance in Figure 8 are 17.3 km and 28.6 km, respectively).

[Figure]

**Figure R1.** Topographic characteristics of the location of ground-based lidar.

The random noise in the ACDL profiles is higher than the ground-based lidar profiles due to their much longer measurement range and shorter integration times, and the daytime noise level is higher than nighttime due to the statistical fluctuations associated with solar background light. Reducing the noise to the required level requires averaging profiles along the ground track (Gimmestad et al., 2017). However, spatial aerosol inhomogeneities introduce systematic error into the averages. The Zhangye and Dunhuang lidar station are located at 1454 m a.s.l. and 1142 m a.s.l., respectively, with a complex topography of the surrounding area (Figure R1) that makes very difficult the comparison in the planetary boundary layer with satellite data acquired with imperfect spatial coincidence. Overall, the observation comparison results between the ACDL and ground-based lidar are acceptable.

In the end of Section 3, we added "***The discrepancies in the comparison between ACDL and ground-based lidar measurement are possible because of the several factors such as different lidar system parameters, the difference in detection distance between spaceborne and ground-based systems and also can be due to the***

*inhomogeneity of the atmosphere (Chiang et al., 2011; Belegante et al., 2018).*
*Overall, the observation comparison results between the ACDL and ground-based*
*lidar are acceptable.*"

Reference:

Chiang, C.-W., Kumar Das, S., Shih, Y.-F., Liao, H.-S., and Nee, J.-B.: Comparison of CALIPSO and ground-based lidar profiles over Chung-Li, Taiwan, Journal of Quantitative Spectroscopy and Radiative Transfer, 112, 197–203, https://doi.org/10.1016/j.jqsrt.2010.05.002, 2011.

Gimmestad, G., Forrister, H., Grigas, T., and O'Dowd, C.: Comparisons of aerosol backscatter using satellite and ground lidars: implications for calibrating and validating spaceborne lidar, Sci Rep, 7, 42337, https://doi.org/10.1038/srep42337, 2017.

3. What is the specific standard deviation represented by the shadow envelope in Figures 5, 8, and 12? If it is the standard deviation of the black solid line and the green solid line in the figure, then it should be a value. If not, please provide specific explanations.

**Response:** We thank you for raising the question. The black and green shadows in the Figures 5, 8, and 12 represent the standard deviation of ACDL and ground-based lidar observations, respectively. The calculation formula is:

$$\sigma = \sqrt{\frac{1}{N}\sum_{i=1}^{N}(X_i - \mu)^2} \quad (1)$$

Where $\sigma$ is the standard deviation, N is the number of selected profiles. For a certain observation height, $\mu$ is the average value of different profiles, $X_i$ is the corresponding observation value. For ACDL, we select the point nearest to the ground-based lidar position in the ground track and took 10 profiles before and after it for average (black solid line in the figures) and standard deviation (black shadow in the figures) calculation, N=21; For ground-based lidar, we select 10 minutes of observations before and after the ACDL overpass time to calculate the average (green solid line in the figures) and standard deviation (green shadow in the figures), N=5.

Indeed, observations corresponding to a height are able to obtain a mean and a standard deviation, and we denote the random noise of the two observations in the form of "mean ± standard deviation", respectively. The random noise (the range of shadows) in the ACDL profiles is much higher than the ground-based lidar profiles due to their much longer measurement range and shorter integration times, and the daytime noise level is higher than nighttime due to the statistical fluctuations associated with solar background light.

4. In Figure 9, there is a significant difference in VDR values between ACDL and CALIPSO at several altitudes (such as below 2.5km). Why does the author still believe in the last paragraph of Section 3.2 that the observation results of ACDL and CALIPSO have good consistency?

**Response:** Thanks for the comment. At low altitudes, the difference of observations is probably due to the distance between the location of ground-based lidar and the ACDL ground track. For this comparison, the nearest distance between the CALIPSO ground track and the ACDL ground track was 69 km (Figure R2). Due to spatiotemporal variations in humidity and aerosols in the lower atmosphere, as well as non-uniformity in the atmosphere near the surface, these observational differences occur near the planetary boundary layer (Kim et al., 2008; Mamouri et al., 2009; Chiang et al., 2011).

[Figure]

**Figure R2.** Topographic characteristics of the location of ground-based lidar (the solid red line represents the ACDL ground track, while the solid blue line represents the CALIPSO ground track).

In addition, the uneven terrain in the northwest region of China can also affect

satellite observations (Figure R2), resulting in significant differences in the comparison between ACDL and CALIPSO near the surface. The Zhangye lidar station is located at 1454 m a.s.l. with a complex topography of the surrounding area that makes very difficult the comparison in the planetary boundary layer with satellite data acquired with imperfect spatial coincidence. Validation of CALIPSO using ground-based lidar has also found observational differences near the surface (Figure R3). However, the main layering characteristics are evident also in the CALIPSO and ACDL profiles, which are similar to those observed by the ground-based lidar.

[Figure]

**Figure R3.** Height profiles of the backscatter coefficients at (a) 532 and (b) 1064 nm, the (c) extinction coefficient and (e) lidar ratio at 532 nm, the (e) backscatter-related Ångström exponent for the wavelength pair 532/1064 nm, and the (f) particle depolarization ratio at 532 nm as measured with BERTHA (red) between 2133 and 2317 UTC on 14 June 2008 and CALIPSO (blue) during an overpass at 1528 UTC on 15 June 2008 about 478 km to the west of Praia, Cape Verde (see Table 3). Thin and thick lines denote unsmoothed and smoothed (660 m) profiles, respectively. Particle depolarization ratio profiles measured with BERTHA are compared to the ones given in the CALIPSO level 2 files (light blue in f) and calculated according to equation (2) (dark blue in Figure 3f). The dotted lines mark the vertical range used for a comparison of the measurements of the two instruments (see column 7 in Table 3) (From Figure 3 of Tesche et al., 2013).

Reference:

Chiang, C.-W., Kumar Das, S., Shih, Y.-F., Liao, H.-S., and Nee, J.-B.: Comparison of CALIPSO and ground-based lidar profiles over Chung-Li, Taiwan, Journal of Quantitative Spectroscopy and Radiative Transfer, 112, 197–203, https://doi.org/10.1016/j.jqsrt.2010.05.002, 2011.

Kim, S.-W., Berthier, S., Raut, J.-C., Chazette, P., Dulac, F., and Yoon, S.-C.: Validation of aerosol and cloud layer structures from the space-borne lidar CALIOP using a ground-based lidar in Seoul, Korea, Atmos. Chem. Phys., 8, 3705–3720, https://doi.org/10.5194/acp-8-3705-2008, 2008.

Mamouri, R.-E., Amiridis, V., Papayannis, A., Giannakaki, E., Tsaknakis, G., and Balis, D.: Validation of CALIPSO space-borne-derived attenuated backscatter coefficient profiles using a ground-based lidar in Athens, Greece, Atmospheric Measurement Techniques, 2, https://doi.org/10.5194/amt-2-513-2009, 2009.

Tesche, M., Wandinger, U., Ansmann, A., Althausen, D., Müller, D., and Omar, A. H.: Ground-based validation of CALIPSO observations of dust and smoke in the Cape Verde region, JGR Atmospheres, 118, 2889–2902, https://doi.org/10.1002/jgrd.50248, 2013.

5. The focus of the paper is to verify the consistency between ACDL and ground-based LiDAR observations, however, the presence of CALIPSO in the abstract and introduction is too strong, which is unnecessary. Although it would be more convincing to verify the consistency between ACDL and CALIPSO, as shown in Figure 9, their consistency is not very ideal.

**Response:** Thanks for your suggestion. At the beginning of the validation, we only used ground-based lidar. When we attempted to submit to the ACP journal, the editor suggested that we consider comparing ACDL with CALIPSO and we followed the editor's suggestion. In our manuscript, our focus is on using ground-based lidar to validate ACDL, and CALIPSO observations are only an auxiliary.

There are differences between ACDL and CALIPSO in comparison, which may be due to differences in satellite parameters (for CALIPSO: field of view (FOV) is 0.13 mrad, the pulse energy is 110 mJ; for ACDL: FOV is 0.2 mrad, the pulse energy is 150

mJ); distance difference between ground tracks. At low altitudes, the difference of observations is probably due to the distance between the location of ground-based lidar and the ACDL ground track. For this comparison, the nearest distance between the CALIPSO ground track and the ACDL ground track was 69 km (Figure R2). Due to spatiotemporal variations in humidity and aerosols in the lower atmosphere, as well as non-uniformity in the atmosphere near the surface, these observational differences occur near the planetary boundary layer (Kim et al., 2008; Mamouri et al., 2009; Chiang et al., 2011). In addition, the uneven terrain in the northwest region of China (Figure R1) can also affect satellite observations, resulting in significant differences in the comparison between ACDL and CALIPSO near the surface. The Zhangye lidar station is located at 1454 m a.s.l. with a complex topography of the surrounding area that makes very difficult the comparison in the planetary boundary layer with satellite data acquired with imperfect spatial coincidence. Similar comparative differences have also been observed in previous studies (Figure R4). However, the main layering characteristics are evident also in the CALIPSO and ACDL profiles, which are similar to those observed by the ground-based lidar.

[Figure]

**Figure R4.** The comparison on vertical profiles of attenuated backscatter measured by CALIPSO (green dash line) and ground-based lidar (red solid line) system on 17 September 2008. Blue dot line is the molecular attenuated backscatter. (For interpretation of the references to colour in this figure legend, the reader is referred to the web version of this article.) (From Figure 2 of Chiang et al., 2011).

Reference:

Chiang, C.-W., Kumar Das, S., Shih, Y.-F., Liao, H.-S., and Nee, J.-B.: Comparison of CALIPSO and ground-based lidar profiles over Chung-Li, Taiwan, Journal of Quantitative Spectroscopy and Radiative Transfer, 112, 197–203, https://doi.org/10.1016/j.jqsrt.2010.05.002, 2011.

Kim, S.-W., Berthier, S., Raut, J.-C., Chazette, P., Dulac, F., and Yoon, S.-C.: Validation of aerosol and cloud layer structures from the space-borne lidar CALIOP using a ground-based lidar in Seoul, Korea, Atmos. Chem. Phys., 8, 3705–3720, https://doi.org/10.5194/acp-8-3705-2008, 2008.

Mamouri, R.-E., Amiridis, V., Papayannis, A., Giannakaki, E., Tsaknakis, G., and Balis, D.: Validation of CALIPSO space-borne-derived attenuated backscatter coefficient profiles using a ground-based lidar in Athens, Greece, Atmospheric Measurement Techniques, 2, https://doi.org/10.5194/amt-2-513-2009, 2009.

6. Similar to the fourth opinion, the difference in observation results between ACDL and ground-based LiDAR in Figure 12 is so significant. Why does the author still believe in the third paragraph of Section 3.3 that both TABC and VDR values are numerically close?

**Response:** Thanks for your suggestion. Figure 12 is a comparison of observations for two cloud cases. Clouds are more complex than aerosols because of their geometrical and optical properties. Especially for spaceborne lidars, the multiple scattering in clouds is not negligible (Winker, 2003). It is well known that spaceborne lidar measurements of ice clouds are typically affected by specular reflection, when observed by lidar pointed near the nadir (Young and Vaughan, 2009). Specular reflection causes anomalously high backscatter (Hogan and Illingworth, 2003). Therefore, many studies have shown that there are differences in spaceborne and ground-based lidar observations when clouds are present (Kim et al., 2008; Mona et al., 2009; Chiang et al., 2011). However, as clouds are a very important atmospheric process, evaluating the recognition ability of newly launched satellites for clouds is equally important. For the most cloud cases, the heights of clouds are different from

several meters to several hundred meters between spaceborne and ground-based lidar measurements (Figure R5). It is due to the various distributions of clouds and the observations are not being made exactly at the same place. Comparison results of clouds illustrate the limitations of spaceborne downward-looking and ground-based upward-looking lidar measurements due to strong signal attenuations, and imply that only information on the cloud top (bottom) height is reliable from satellite-based ACDL (ground-based lidar) observations. However, the complementarity between space-borne and ground-based lidar observations can provide complete vertical structures of aerosols and clouds.

We added "***Due to the inhomogeneous horizontal distribution of aerosols in the lower troposphere and clouds, there are some differences between cloud and boundary layer lidar measurements.***" in Section 4. And we changed sentence "However, numerically both TABC and VDR values are closer" to "***However, both observations show similar cloud structures.***" in Section 3.3.

[Figure]

**Figure R5.** Vertical profiles of (a) CALIOP-derived and (b) SNU lidar-derived total attenuated backscatter at 532 nm wavelength, and (c) apparent scattering ratios $R_{app}$ at 532 nm calculated from the CALIOP (green lines) and the SNU lidar (blue line) measurements on 14 September 2006 (17:41 UTC) (From Figure 6 of Kim et al., 2008).

Reference:

Chiang, C.-W., Kumar Das, S., Shih, Y.-F., Liao, H.-S., and Nee, J.-B.: Comparison of CALIPSO and ground-based lidar profiles over Chung-Li, Taiwan, Journal of Quantitative Spectroscopy and Radiative Transfer, 112, 197–203, https://doi.org/10.1016/j.jqsrt.2010.05.002, 2011.

Hogan, R. J. and Illingworth, A. J.: The effect of specular reflection on spaceborne lidar measurements of ice clouds, Report of the ESA Retrieval algorithm for

EarthCARE project, 2003.

Kim, S.-W., Berthier, S., Raut, J.-C., Chazette, P., Dulac, F., and Yoon, S.-C.: Validation of aerosol and cloud layer structures from the space-borne lidar CALIOP using a ground-based lidar in Seoul, Korea, Atmos. Chem. Phys., 8, 3705–3720, https://doi.org/10.5194/acp-8-3705-2008, 2008.

Mona, L., Pappalardo, G., Amodeo, A., D'Amico, G., Madonna, F., Boselli, A., Giunta, A., Russo, F., and Cuomo, V.: One year of CNR-IMAA multi-wavelength Raman lidar measurements in coincidence with CALIPSO overpasses: Level 1 products comparison, Atmos. Chem. Phys., 9, 7213–7228, https://doi.org/10.5194/acp-9-7213-2009, 2009.

Winker, D.: Accounting for multiple scattering in retrievals from space lidar, Proc. SPIE Int. Soc. Opt. Eng., 5059, 128–139, 2003.

Young, S. A. and Vaughan, M. A.: The Retrieval of Profiles of Particulate Extinction from Cloud-Aerosol Lidar Infrared Pathfinder Satellite Observations (CALIPSO) Data: Algorithm Description. J. Atmos. Oceanic Technol., 26(6), 1105–1119, 2009.

---

## Author Response (AR2)

**Response to the comments of Reviewers**

**RC1: 'Comment on amt-2023-235', Anonymous Referee #1:**

Thanks for the authors to address my comments. I think the paper is suitable for publicaiton in AMT.

**Response:** We are very much appreciated for your scrutiny of our manuscript, and thanks for your comments.

**RC2: 'Comment on amt-2023-235', Anonymous Referee #3**

For a newly launched space-borne lidar, verifying its observations under different weather conditions can help improve its observation quality and serve the provision of subsequent data retrieval products. The manuscript uses the Belt and Road lidar network and CALIPSO satellite to verify TABC, VDR, extinction coefficient and lidar ratio obtained from the world's first space-borne high spectral resolution lidar ACDL initial observation, showing the good observation performance of ACDL under different typical weather conditions. The topic is of sufficient interest to the communities of study of laser remote sensing and atmospheric aerosol. In general, I find this manuscript to be of interest for publication and appropriate for Atmospheric Measurement Techniques. There are several suggestions for improvement listed below that should be considered by the authors and the editors before publication.

**Response:** Thank you very much for the appreciation of the work.

1. The title of the manuscript uses "space-borne". It is suggested to unify the "satellite-based" in the manuscript to "space-borne".

**Response:** Thank you very much for raising this concern. We have unified the wording throughout the entire manuscript.

2. The abbreviation (AEMS) in line 33 is not mentioned in the following text and can be deleted here.

**Response:** Thanks for your suggestion. We have deleted it.

3. In line 53, the "Observations" should be "Observation".

**Response:** Thank you very much for your suggestion. We have corrected it.

4. The second paragraph of the introduction describes the related validation work of the CALIPSO satellite, and it is recommended to describe it in chronological order according to the publication of the article.

**Response:** Thanks. The introduction part has been revised. Please see the lines 54-58.

5. Please double check the use of verb tenses in the introduction section to ensure consistency in the tense of the manuscript.

**Response:** Thank you very much for pointing out this mistake. We have standardized the verb tenses in the sentences.

6. In line 76, "section" should be "Section".

**Response:** Thanks. We have corrected it.

7. In line 78, "ground-based lidar network" should use "BR-lidarnet".

**Response:** Thanks. We have corrected it.

8. "In addition" in line 86 is repeated with "Additionally" in the previous sentence. It is recommended to change the word.

**Response:** Thanks. We have replaced it with "***Moreover***".

9. Regarding ACDL, ground-based lidar, and CALIPSO in Section 2, additional parameters for the lidar system could be added as appropriate. This can allow readers to see the differences between instruments more intuitively.

**Response:** Thank you very much for your suggestion. We have added more detailed parameter descriptions in lines 89-90 and lines 106-107.

10. It is suggested to unify the "background subtraction" in line 106 with Figure 2 and change it to "background correction".

**Response:** Thanks. We have revised it.

11. At the beginning of Section 2.3, the full name "The Cloud Aerosol Lidar and Infrared Pathfinder Satellite Observation (CALIPSO)" is already written in the introduction, and abbreviations can be used directly here.

**Response:** Thanks. We have revised it.

12. The description of the observation time for the selected cases in lines 198 and 190 is redundant. It is recommended to delete it appropriately to simplify the expression. And one of the cloud cases compared is at night. Please verify in detail.

**Response:** Thanks. Sentence has been revised with better clarity.

13. In line 199, check that the tense of the sentence is consistent with the following text.

**Response:** Thank you very much. Sentence has been revised with better clarity.

14. In line 201, "Meanwhile, the air quality index (AQI) values of these two days in order are 43 and 45" should change to "Meanwhile, the air quality index (AQI) values of these two days are 43 and 45, respectively". Similarly, in line 206, add ", respectively" at the end of the sentence.

**Response:** Thanks. We have revised it.

15. In line 232, please specify which "ground stations" it is.

**Response:** Thank you very much for your suggestion. We have revised it as "*Zhangye site*".

16. The tense of the sentence in line 246 should be consistent with the following text.

**Response:** Thank you very much. We have revised it.

17. In line 253, "The historical weather forecast also shows a cloudy condition over Zhangye on that day." It is better to change "forecast" to "record" here, because what has already happened cannot be called "forecast" anymore.

**Response:** Thank you very much for your suggestion. We have revised it.

18. Please double check the entire text, there should be a space between the unit and the number.

**Response:** Thank you very much. We have revised it.

19. The content starting from line 274 is different from the analysis of cloud cases, and focuses on studying the observation consistency between ACDL and ground-based lidar from a statistical perspective. Therefore, it is recommended to separate a section for discussion.

**Response:** Thank you very much for your suggestion. We have added a new section "*3.4 Comparison of ACDL and ground-based lidar observations*" to discuss the observation consistency between ACDL and ground-based lidar.

20. In line 274, the "affects" should be "affect".

**Response:** Thank you very much. We have revised it.

21. The descriptions in lines 289 and 292 regarding the selection of different weather conditions for individual cases are duplicated. Please revise these two sentences.

**Response:** Thank you very much. We have revised it.

22. The tense of the sentence on line 301 should be consistent with the previous text.

**Response:** Thanks. We have revised it.

23. In line 309, the "high spectral lidar" should be written as "HSRL".

**Response:** Thank you very much. We have revised it.